# Biomarkers for the Prediction and Judgement of Sepsis and Sepsis Complications: A Step towards *precision medicine*?

**DOI:** 10.3390/jcm11195782

**Published:** 2022-09-29

**Authors:** Thilo von Groote, Melanie Meersch-Dini

**Affiliations:** Department of Anaesthesiology, Intensive Care and Pain Medicine, University Hospital Muenster, D-48149 Muenster, Germany

**Keywords:** sepsis, biomarkers, *precision medicine*

## Abstract

Sepsis and septic shock are a major public health concern and are still associated with high rates of morbidity and mortality. Whilst there is growing understanding of different phenotypes and endotypes of sepsis, all too often treatment strategies still only employ a “one-size-fits-all” approach. Biomarkers offer a unique opportunity to close this gap to more precise treatment approaches by providing insight into clinically hidden, yet complex, pathophysiology, or by individualizing treatment pathways. Predicting and evaluating systemic inflammation, sepsis or septic shock are essential to improve outcomes for these patients. Besides opportunities to improve patient care, employing biomarkers offers a unique opportunity to improve clinical research in patients with sepsis. The high rate of negative clinical trials in this field may partly be explained by a high degree of heterogeneity in patient cohorts and a lack of understanding of specific endotypes or phenotypes. Moving forward, biomarkers can support the selection of more homogeneous cohorts, thereby potentially improving study conditions of clinical trials. This may finally pave the way to a *precision medicine* approach to sepsis, septic shock and complication of sepsis in the future.

## 1. Introduction

Systemic inflammation, sepsis and septic shock are a complex continuum that is characterized by an inadequate systemic immune response to an initial stimulus.

Dysregulated systemic inflammation due to an underlying infection may cause the clinical syndrome of sepsis or septic shock, defined as a “life-threatening organ dysfunction due to a dysregulated host response to infection” in the Sepsis-3 definition [1]. The exact epidemiology of sepsis or septic shock remains unclear; however, it is clear that this is a major public health issue worldwide, with an estimated 31.5 million cases of sepsis annually and 5.3 million attributed deaths globally per year [2,3,4]. Large epidemiological studies in high-income countries suggest a dramatically rising number of patients with sepsis [5,6,7]. Severe forms of sepsis can cause shock by dysregulation of macro- and microvascular hemodynamics through various mechanisms. Organ failure such as acute kidney injury (AKI) are common complications of sepsis, further increasing complication rates and mortality [8]. Early diagnosis and treatment of sepsis is therefore essential, as delayed treatment initiation is associated with an increased risk of mortality [9]. However, the early detection of patients with sepsis by clinical diagnosis remains difficult and often delays the timely start of treatment. Finally, the wide range of causes and pathophysiology of sepsis further complicates clinical judgment.

Given the complexity and severity of sepsis and septic shock, a more *precision medicine*-oriented approach is urgently required.

*Precision medicine* has become a catchphrase and receives much attention. It has widely replaced the term “*personalized medicine*” over criticism that this implies that physicians did not always treat patients in a personalized manner. Specifically, *precision medicine* is about matching treatment approaches as closely as possible to the patients’ unique individual characteristics, based on biological, genetic, clinical or other patient data, and to obtain such data as exactly as possible [10,11]. Whilst this is surely not a new concept, the opportunities for employing this approach have grown significantly in recent years based on the drastic increase in available data per patient. This is further supported by innovative methods to easily obtain data with high granularity, for example, by using point-of-care biomarker measurements, and technology to further analyze these data with sophisticated methods—for example, employing artificial intelligence and deep-learning-based phenotyping. Whilst there is no official definition of *precision medicine*, it may be defined in several ways:

Firstly, focusing on the treatment approach, *precision medicine* employs therapies that are as specific as possible to an identified pathological mechanism of action. As an example, the availability and use of targeted antibody therapy illustrates this aspect. Secondly, *precision medicine* can be defined based on the data used and methods by which these data were acquired. Innovative methods offer a more detailed understanding of a pathological process in an individual patient. Extensive profiling of genetic, proteomic or metabolomic characteristics, as well as the use of specific biomarkers, are only some methods used to stratify patients in ways that allow for more individualized diagnosis, prognostication or treatment. Agusti et al. described this approach as the quest to identify “treatable traits” in contrast to the classic “signs-and-symptoms” approach of clinical medicine [12].

However, implementing *precision medicine* approaches in routine clinical practice will increase complexity and potentially costs. Furthermore, the lack of data on diverse patient populations, but also the limited understandings of gender, race or other social determinants of health, limit the generalized implementability of *precision medicine* approaches. Translating the results of *precision medicine* research into routine practice must overcome these barriers, and *precision medicine* approaches must ensure an equitable impact on the target populations [13]. This implies significant ethical considerations and requires special mindfulness in this regard. Contributing to this issue is the uncertainty of the degree of required evidence in order to introduce a new *precision medicine* test or procedure. Determining required evidence thresholds of estimated benefit, risk for harm, potential disparities in access or development of *precision medicine* approaches are therefore warranted to guide the clinical uptake of these strategies [14]. For example, the case of anticoagulation therapy illustrates this dilemma: whilst a large body of evidence suggests important pharmacogenetic factors and large interindividual variability of treatment response to anticoagulation drugs such as warfarin, the evidence for *precision medicine* anticoagulation therapy to improve outcomes remains very controversial [15]. Finally, patients must be empowered to understand this new paradigm in medical practice and be able to provide or decline informed consent [16].

## 2. What Are Biomarkers?

Biomarkers are measurable characteristics that can provide insight into biological or pathological processes [17]. Biomarkers can be categorized into diagnostic, monitoring, prognostic and stratification biomarkers [18]. While diagnostic biomarkers support the early detection, exact diagnosis and eventual identification of underlying pathophysiology (endotype), monitoring biomarkers can provide information on treatment response and effect, as well as residual disease activity. Prognostic parameters may allow prognostication or support prognostic models. Finally, stratification biomarkers allow for subclassification or staging of diseases, guided by disease severity, underlying pathophysiological mechanism or predicted outcome [19].

## 3. The Role of Biomarkers in *precision medicine*

To improve this status quo, biomarkers are a promising tool to make *precision medicine* a reality in the management of patients with sepsis or septic shock.

Biomarkers may support caregivers to employ *precision medicine* in sepsis or septic shock through various ways (Figure 1). First, to detect sepsis or septic shock before clinical symptoms or deterioration occurs, and subsequently provide information on the current state of the systemic inflammation or shock. Second, to differentiate distinct causes or phenotypes that may respond differently to therapy. Third, to guide treatment length and intensity. This is important to limit negative side effects, such as antibiotic resistance due to unnecessary long antibiotic treatment. Fourth, biomarkers can support physicians by predicting or detecting complications of sepsis. Fifth, biomarkers can improve prognostication and therefore support clinical management decisions.

## 4. Will Biomarkers Pave the Way for *precision medicine* Sepsis Trials?

Many clinical trials in the field of intensive care medicine have reported negative or conflicting results. The field of critical care medicine, especially concerning inflammatory syndromes such as sepsis or ARDS, is afflicted with high numbers of negative trials for which several possible reasons have been identified [20]. Some of these are the complexity and multimorbidity of patient cohorts with high rates of mortality as well as the often unclear pathophysiology. This is aggravated by difficulties in recruiting large numbers of patients to detect smaller treatment effects as well as difficulties in optimal timing and end point selection [21,22]. High levels of heterogeneity are of special importance in critical care and sepsis research [23]. Clinical criteria currently used for the diagnosis of sepsis are neither able to adequately delineate patients with different phenotypes nor to predict who will benefit from specific therapies. Instead of using these unspecific syndrome groups, more precise and elaborate characterization of patient cohorts in the clinical research of sepsis or septic shock is required. Recent advances in biotechnology and artificial intelligence have significantly improved the understanding of clinical, molecular as well as genetic mechanisms underlying or influencing the development and progress of sepsis. Furthermore, studies have also demonstrated the different treatment response and outcomes of distinct phenotypes, such as in sepsis-induced AKI [24].

Many look at the field of oncology as an example of the advanced application of personalized medicine, both in research and clinical practice [25]. This personalized approach, using more granular data and biomarkers for patient selection and treatment allocation, enables researchers in the field of oncology to select more homogenous patient cohorts that are more likely to respond to specific treatments. Ideally, biomarkers are used that identify “treatable traits” such as genetic mutations or pathophysiologic derangements, for which specific therapies exist.

In sepsis, biomarkers have a key role to identify these specific subgroups and support evaluation of a patient’s individual and current inflammatory response (Figure 2).

Clinical trials of sepsis have accordingly taken up biomarker-based “enrichment-strategies” [26,27]. For example, Meyer et al. developed a genetic classification of variants of the interleukin-1 receptor antagonist-associated genes, offering a possible enrichment strategy for the use of interleukin-1-receptor antibodies in future clinical trials and practice [28,29].

Every year, numerous novel biomarkers of sepsis or septic shock are published, but this number has been slightly declining in recent years [30]. However, of the numerous studies, only few have investigated the effect of sepsis biomarkers on treatment decisions or clinical outcomes when these biomarkers are integrated into clinical pathways. In this review, we will describe three use cases in which biomarkers can make *precision medicine* a reality in the field of sepsis: First, we will review the role of biomarkers for the judgment of inflammatory state and biomarker-guided or -targeted immunomodulatory therapies of systemic inflammation leading to sepsis. Second, the example of adrenomedullin (ADM)-guided application of adrecizumab in septic shock will demonstrate how biomarkers can guide specific therapies that are part of complex and dynamic clinical syndromes such as septic shock. Third, we illustrate how biomarkers can improve the clinical management of severe complications of sepsis, using the example of sepsis-associated acute kidney injury (sAKI).

## 5. Three Exemplary Use Cases for Biomarkers in the Context of Sepsis

### 5.1. Use Case No. 1: Biomarker-Guided Evaluation and Therapy of Patients with Dysregulated Systemic Inflammation and Sepsis: Implementing a precision medicine Approach

Many clinical trials have investigated therapies that target immune response; however, most of these trials have been negative trials. Given that most of the trials performed so far have applied immunomodulatory treatments to rather broad, unselected cohorts of patients with clinical syndromes—such as systemic inflammatory response syndrome (SIRS), sepsis, or septic shock—a more *precision medicine*-oriented approach is urgently required [31]. For example, broad and undifferentiated immunosuppressive treatment with corticosteroids remains controversial, and several large-scale clinical trials have reported conflicting results regarding effects on clinical outcomes [32,33,34]. Most initial trials investigating the effectiveness of anti-inflammatory antibodies in broad cohorts of sepsis patients did not demonstrate significant improvements in clinical outcomes [35,36]. Performing trials of immunosuppressive treatments on unselected cohorts of patients will likely result in the application of these treatments to a heterogeneous cohort of patients with both hyper- and hypoinflammatory states. However, patients in the phase of immunoparalysis will likely have little or no benefit from further therapeutic immunosuppression, or this may even cause harm.

Despite growing understanding of these pathophysiological immune processes in sepsis, therapeutic targeting of cytokines and targeted immunosuppression remains a difficult challenge [37].

Numerous biomarkers have been studied for potential applications in systemic inflammation, and Table 1 provides an overview of selected biomarkers for the precision-medicine-oriented management of sepsis patients.

Implementing biomarker-guided treatment algorithms could improve response-to-treatment rates significantly and enable a *precision medicine* approach, even in complex circumstances of sepsis or critical illness.

### 5.2. Use Case No. 2: Specific Biomarkers for Specific Therapies

Antibody-treatments target specific pathophysiologic pathways and offer new tools in the treatment of critically ill patients with septic shock. However, such antibody therapies are costly and require careful selection of suitable patients. Specific biomarkers that identify patients with potential treatment benefits from these therapies are therefore required.

In progressive and severe sepsis, disintegration of the endothelial barrier is a key pathophysiologic component of septic shock and drives vascular leakage, tissue oedema and hypotension [99]. In healthy states, peptide hormones such as adrenomedullin (ADM) stabilize and regulate the endothelial barrier, but are disturbed in sepsis and septic shock. Therefore, elevated ADM levels in the blood provide insight into the clinical state of patients with this endotype of septic shock. With increasing damage of the endothelial barrier, ADM furthermore leaks out of the blood vessels, potentially initiating a *circulus vitiosus* of further endothelial damage due to ADM dysbalance.

With innovative and precise antibody treatments such as Adrecizumab, a positive biomarker (in this case ADM) can result in direct therapeutic consequences. Adrecizumab is a monoclonal antibody that binds directly to intravascular ADM, thereby preventing ADM leakage into the extravascular space [100]. Additionally, contributing to the desired effect, this Adrecizumab–ADM-complex results in a longer intravascular half-life of ADM, with longer functional activity to stabilize the vascular endothelium. Several studies in rats and murine models suggest a potential survival benefit and decreased vascular leakage and have resulted in the conduct of first in-human studies of Adrecizumab [101,102]. In a first-in-human study of experimental endotoxemia in healthy subjects, Adrecizumab effectively increased the intravascular levels of ADM, and no safety concerns were observed [103]. Following up on these results, Laterre et al. conducted the AdrenOSS-2 phase 2a trial to investigate a biomarker-guided approach to Adrecizumab treatment in patients with septic shock and demonstrated a favorable safety profile and tolerability. Whilst the trial suggested a potential 28-day mortality benefit, these results did not reach statistical significance [104]. However, this phase 2a study was not powered to assess this outcome. Further research is required to investigate the effectiveness of such promising biomarker-guided treatments and their potential impact on clinical outcomes for patients with septic shock based on this endotype of vascular damage.

### 5.3. Use Case No. 3: Biomarker-Guided precision medicine to Treat Sepsis Complications: Sepsis-Associated Acute Kidney Injury (sAKI)

In our final use case, we argue that biomarkers can introduce a *precision medicine* approach to improve clinical trial design and the management of sepsis complications, such as sepsis-associated acute kidney injury (AKI). AKI is a common and often severe complication of sepsis and septic shock [105,106]. Different pathophysiological mechanisms, such as renal hypoperfusion, endothelial leakage, microcirculatory dysfunction, circulating toxins and other risk factors, are present and lead to sepsis-associated AKI (sAKI) [107,108,109,110]. If sAKI occurs, this is associated with excessive rates of morbidity and mortality [111,112]. Still, no causative treatment for sAKI exists; therefore, optimal preventive strategies and supportive treatments are essential. Biomarkers of AKI can enable clinicians to identify patients that might benefit from specific interventions and tailor therapy to individual needs [113].

Several biomarkers of AKI have been studied, and tissue inhibitor of metalloproteinase-2 and insulin such as growth factor binding protein 7 [TIMP-2]*[IGFBP7] are among the most established biomarkers [114,115]. [TIMP2] and [IGFBP7] are biomarkers of cell-cycle arrest that detect subclinical stages of AKI. In non-septic AKI, high-quality evidence supports a [TIMP-2]*[IGFBP7] biomarker-guided treatment approach. The PrevAKI-trials have demonstrated in a cardiac surgery cohort that a biomarker-guided patient selection and treatment initiation effectively reduces the rates and severity of AKI [116,117]. Similarly, Göcze et al. reported reduced severity of AKI in a single-center study of non-cardiac patients after major surgery, also using [TIMP-2]*[IGFBP7] to identify patients at high risk for AKI and to initiate preventive treatments [118]. A definitive, multicenter trial to investigate this effect in patients after major surgery is currently being performed in Europe [119]. A trial investigating such a biomarker-guided approach for the treatment of sAKI is on the way. Putting AKI into the context of sepsis, this and other biomarkers offer great potential to improve research and clinical care for these patients.

Moving even one step further, combining biomarker-based endotyping with clinical subphenotyping, Bhatraju et al. employed latent class analysis methodology and parsimonious classification models to investigate distinct subphenotypes of sAKI [120]. They identified two distinct subphenotypes of sAKI that were associated with significantly different survival rates and renal recovery. These distinct sAKI subphenotypes did not differ in epidemiological characteristics, but did significantly differ in clinical phenotype, markers of endothelial dysfunction as well as inflammatory markers. Interestingly, the plasma ratio between Ang-2/Ang-1 and sTNFR-1 adequately differentiated the two subphenotypes. Endothelial growth factors, such as Ang-1 and Ang-2, regulate endothelial integrity and safeguard the physiologic balance between stability and renewal of vascular endothelium. A dysbalanced ratio of these opposing factors may represent an AKI endotype in which endothelial dysfunction may play a major role. In this study, this was associated with significantly worse outcomes in patients with sepsis and AKI [121,122]. In a final part of this study, the authors applied this biomarker-driven classification model to an external replication cohort of the Vasopressin and Septic Shock Trial (VASST) with an astonishing result: patients in one subphenotype had a strong benefit from early addition of vasopressin to norepinephrine, but patients in the other subphenotype did not have this benefit [123]. This study demonstrated an example of how a *precision medicine* approach by use of biomarkers can help identify treatment responders. Such approaches could pave the way to avoid the vast majority of negative sepsis trials, given the imprecise application of study interventions to undefined study populations.

## 6. Outlook: The Coming Era of—Omics Technology?

Moving forward, adding further genetic, as well as proteomic, metabolomic or transcriptomic information, could theoretically lead to even better prognostication or the choice of more adequate treatments; however, this requires demonstration of improved clinical outcomes in further clinical studies [124,125]. The recent developments in omics technology yield further hope for a more detailed understanding of disease pathophysiology in individual patients. This could further drive *precision medicine* in sepsis and could support biomarker development and validation [126,127]. Additionally, more agnostic tissue interrogation approaches, such as single-cell or single-nuclei RNA sequencing technology, could further support these biomarker-driven approaches by providing further insight into the injury mechanism and localization of injury [128]. These approaches could bring the granularity needed for a true *precision medicine* treatment approach to sepsis complications, such as AKI. Finally, genetic information may also play a bigger role in the future. For example, Davenport et al. used whole genome expression profiling for septic shock endotyping and reported that the genomic landscape had significant implications for the individual host response and clinical outcomes [127].

Several studies suggested that a more sophisticated characterization of a patient’s individual genetic, transcriptomic, immune and clinical profile is predictive of response to treatment [129,130,131]. Using omics technology, Sweeney et al. identified and externally validated three very distinct subtypes of sepsis in a large multicenter dataset. These subtypes of sepsis were labelled as “Inflammopathic”, “Coagulopathic” and “Adaptive”. However, any omics technology also faces limitations, such as interference of human DNA, amplification biases and the need for the effective lysis of all target microbes. Finally, they are still often costly and require expert personnel for their conduction. These technologies currently remain rather experimental and are not yet ready to replace blood tests and biomarkers of sepsis. The value of genetic profiling and ways to integrate this into clinical management strategies requires further evaluation with regard to *precision medicine* in the context of sepsis.

## 7. Conclusions

Sepsis and septic shock are heterogeneous clinical syndromes, and multiple immunological and pathophysiological endotypes exist. Hence, a *precision medicine* approach is urgently required to move the care of patients with sepsis and septic shock to the next level—moving away from a “one-size-fits-all” approach. Rather, individual assessment of distinct endotypes and phenotypes is necessary to guide specific therapies and the management of sepsis complications. Biomarkers are a key tool to achieve this goal. Biomarkers can detect sepsis early and provide information on the current state of the dysregulated systemic inflammation. Furthermore, biomarkers can differentiate endotypes or phenotypes that may even respond differently to therapy and can guide such treatment. Finally, biomarkers can predict or detect sepsis complications, as well as improve prognostication and therefore support clinical management decisions. Further translational and clinical research is required that investigates how the integration of biomarkers into treatment pathways can improve patient outcomes in sepsis or septic shock and complications of such. Biomarkers will enable remarkably improved patient selection in clinical trials, allowing for the recruitment of more homogenous patient cohorts.

## Figures and Tables

**Figure 1 jcm-11-05782-f001:**
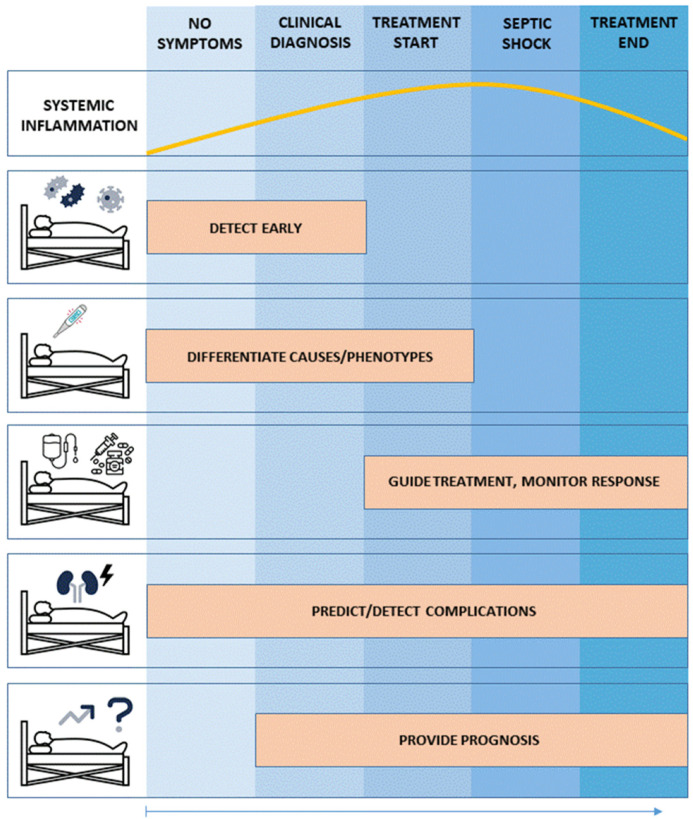
Applications of biomarkers in the clinical pathway of sepsis.

**Figure 2 jcm-11-05782-f002:**
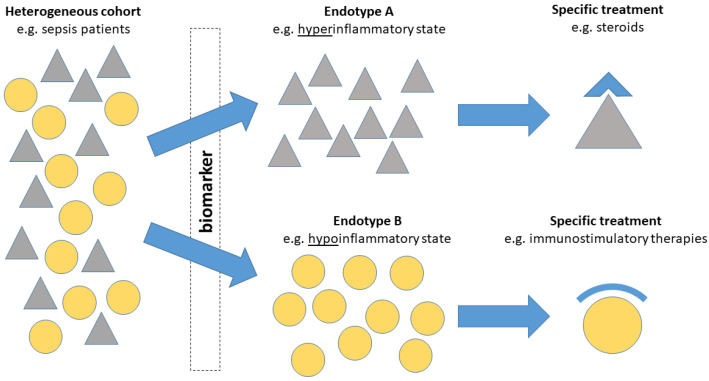
Biomarker-enrichment approach (compare Wong et al. [10]).

**Table 1 jcm-11-05782-t001:** Biomarkers with potential application in sepsis or septic shock.

Immunological Biomarkers	
**C-reactive protein (CRP)**	Indicates acute systemic inflammation [38] Screening for early onset neonatal sepsis [39] (Predict survival in patients with sepsis) [40]
**Procalcitonin (PCT)**	Diagnosis of sepsis [41,42] Suggest bacterial infection [43] Monitor treatment response to antibiotics and guide cessation of antibiotic treatment [44,45,46,47]
**Presepsin (soluble CD14)**	Early detection of sepsis (earlier increase than PCT and CRP) [48] Monitor host response [49] Higher in patients with bacterial infection [50] May be combined with other biomarkers in a panel [51] No validity in patients with acute kidney injury [52]
**Interleukin-6 (IL-6)**	Early detection of sepsis [53,54] Early detection of SIRS [55] Differentiate infectious from sterile SIRS [56]
**Interleukin-8 (IL-8)**	Diagnosis of sepsis [57]
**CD64 expression on neutrophils (nCD64)**	Early detection of sepsis [58,59,60,61,62,63,64] (monitoring of sepsis) [63,65] Prognostic marker of sepsis [66,67]
**Soluble programmed death ligand 1 (sPD-L1)**	Detect immunosuppressed states in sepsis patients [68,69] Potential therapeutic target [70]
**HLA-DR expression on antigen-presenting cells**	High HLA-DR expression: Detect hyperinflammatory state [71] Low HLA-DR expression: Detect immunosuppressed state [72] Predict poor survival in patients with septic shock [73] Potential biomarker for enrichment of clinical trials of sepsis [74]
**Pentraxin (PTX-3)**	Assessment of septic shock severity [75,76] Prediction of mortality in patients with sepsis or septic shock [77]
**Complement protein 5a (C5a)**	Limited utility in sepsis due to both pro- and anti-inflammatory effects [78,79]
** Infectious biomarkers **	
**Lipopolysaccharide-binding protein (LPS-bp)**	Discriminate sterile from infectious basis of SIRS or sepsis [80]
**Pathogen-associated molecular patterns (PAMPs)**	Early detection of pathogen-based immune stimuli [81,82,83,84]
** Biomarkers of endothelial or glycocalyx dysfunction **	
**Syndecan-1**	Assessment of endothelial barrier dysfunction in sepsis [85] Predict organ failure due to endothelial dysfunction [86] Prediction of DIC or coagulatory dysfunction in sepsis-associated endothelial dysfunction [87,88,89] (May be helpful to guide fluid resuscitation in early sepsis) [90]
**Adrenomedullin (ADM)**	s. below (use case no. 2)
**Angiopoietin-1, -2**	Detect fluid overload and endothelial leakage in sepsis [91,92] Predict septic shock [93]
**Thrombomodulin**	Predict multi organ failure and DIC in sepsis [94,95]
**Heparanase-1 and -2 (Hpa-1, Hpa-2)**	May identify septic patients with potential benefit from therapeutic plasma exchange therapy [96] Potential therapeutic target [97,98]

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
