# Peer review of "Biomarkers for the Prediction and Judgement of Sepsis and Sepsis Complications: A Step towards precision medicine?"

_jcm, 2022, doi:10.3390/jcm11195782_

Round 1

Reviewer 1 Report

Overall, the manuscript is well written and comprehensive. However, the authors provided insufficient evidence for the lack of certain treatment effects. Negative trials may happen for numerous reasons, only attributing it to the presence of a heterogenous patient population without evidence is inadequate. The manuscript could be shortened.

Author Response

  1. Overall, the manuscript is well written and comprehensive. However, the authors provided insufficient evidence for the lack of certain treatment effects. Negative trials may happen for numerous reasons, only attributing it to the presence of a heterogenous patient population without evidence is inadequate.

RESPONSE: This is an important point and we agree that this may have not been sufficiently elaborated on in the initial version of the manuscript. Therefore, we have added more information and thoughts in this regard to the manuscript (page 2, lines 51-53; page 3, lines 128-135). However, we would like to argue that high degree of heterogeneity is considered a key aspect of negative trials in critical care medicine and especially sepsis/septic shock and our review therefore focusses on this aspect.

  1. The manuscript could be shortened.

RESPONSE: We shortened the manuscript (see revised manuscript).

Reviewer 2 Report

Dear authors,

Thank you for this manuscript dealing with sepsis biomarkers and their usefulness in sepsis and septic shock. There is indeed an urgent need to review the methodologies of the studies performed.

The authors give their vision about the role of biomarkers in sepsis and this vision could be more general in the use of a biomarker in a defined pathological state. They give their approach with practical examples. However, this view is not new and it is well known that approaches to precision medicine can lead, paradoxically, to increased levels of uncertainty.

Minor points : 

-        The authors' arguments about precision medicine are not sufficiently counterbalanced by arguments that do not justify the use of precision medicine. There are numerous case examples in the literature describing that this approach does not work as well as current standard approaches, for instance for oral anticoagulants. Please adapt the manuscript accordingly.

- The uncertainty described above should be discussed in the manuscript.

-        Figure 1: A very schematic figure, but the symbols in the left-hand column are not sufficiently understandable. What is the interest for the reader of such a schematization? Specify these symbols with words (diagnosis, monitoring,…) 

Author Response

  1. Dear authors, Thank you for this manuscript dealing with sepsis biomarkers and their usefulness in sepsis and septic shock. There is indeed an urgent need to review the methodologies of the studies performed. The authors give their vision about the role of biomarkers in sepsis and this vision could be more general in the use of a biomarker in a defined pathological state. They give their approach with practical examples. However, this view is not new and it is well known that approaches to precision medicine can lead, paradoxically, to increased levels of uncertainty.

RESPONSE: We agree with the reviewer but this review aims at summarizing existing concepts in a comprehensive manner and put forward several use cases to illustrate innovative current or future directions of a biomarker-based precision medicine approach. Indeed, the practical implementation of precision medicine may lead to increased levels of complexity, costs, and levels of uncertainty. Furthermore, the introduction of precision medicine approaches is associated with important ethical and societal questions. We wrote the following:

“However, implementing precision medicine approaches in routine clinical practice will increase complexity and potentially costs. Furthermore, the lack of data on diverse patient populations, but also the limited understandings of gender, race or other social determinants of health, limit the generalized implementability of precision medicine approaches. Translating the results of precision medicine research into routine practice must overcome these barriers and precision medicine approaches must ensure an equitable impact on the target populations[i]. This implies significant ethical considerations and requires special mindfulness in this regard. Contributing to this issue is the uncertainty of degree of required evidence in order to introduce a new precision medicine test or procedure. Determining required evidence thresholds of estimated benefit, risk for harm, potential disparities in access or development of precision medicine approaches are therefore warranted to guide clinical uptake of these strategies[ii]. For example, the case of anticoagulation therapy illustrates this dilemma: Whilst a large body of evidence suggests important pharmacogenetic factors and large inter-individual variability of treatment response to anticoagulation drugs like warfarin, the evidence for precision medicine anticoagulation therapy to improve outcomes remains very controversial[iii]. Finally, patients must be empowered to understand this new paradigm in medical practice and be able to provide or decline informed consent[iv].” (page 2, lines 79-96).

  1. Minor points: The authors' arguments about precision medicine are not sufficiently counterbalanced by arguments that do not justify the use of precision medicine. There are numerous case examples in the literature describing that this approach does not work as well as current standard approaches, for instance for oral anticoagulants. Please adapt the manuscript accordingly

RESPONSE: Again, thank you for pointing this out. We have integrated this in the manuscript (see above, Page 2, lines 79-96).

  1. The uncertainty described above should be discussed in the manuscript.

RESPONSE: We agree with the reviewer and refer to our response above (2nd response) about how this was revised and addressed.

  1. Figure 1: A very schematic figure, but the symbols in the left-hand column are not sufficiently understandable. What is the interest for the reader of such a schematization? Specify these symbols with words (diagnosis, monitoring,…)

RESPONSE: Thank you for pointing this out. The figure aims to provide oversight of the different time points and timeframes of biomarkers in sepsis/septic shock, as well as possible indications. We followed the reviewer’s kind advice and added several text fields to the symbols to improve readability and avoid loss of information or clarity due to schematization.

Round 2

Reviewer 1 Report

No further comments. The authors have addressed my comments.